# Impact of an acceptance facilitating intervention on psychotherapists' acceptance of blended therapy

Harald Baumeister[1]*, Yannik Terhorst[1], Cora Grässle[1], Maren Freudenstein[1], Rüdiger Nübling[2], David Daniel Ebert[3]

1 Department of Clinical Psychology and Psychotherapy, Institute of Psychology and Education, Ulm University, Ulm, Germany, 2 Chamber of Psychotherapists Baden-Württemberg, Stuttgart, Germany, 3 Faculty of Behavioural and Movement Sciences, Clinical, Neuro- & Developmental Psychology, VU Amsterdam, Amsterdam, The Netherlands

* harald.baumeister@uni-ulm.de

**Data Availability Statement:** All relevant data are within the manuscript and its Supporting Information files.

## Abstract

Blended therapy is a new approach combining advantages of face-to-face psychotherapy and Internet- and mobile-based interventions. Acceptance is a fundamental precondition for its implementation. The aim of this study was to assess 1) the acceptance of psychotherapists towards blended therapy, 2) the effectiveness of an acceptance facilitating intervention (AFI) on psychotherapists' acceptance towards blended therapy and 3) to identify potential effect moderators. Psychotherapists (N = 284) were randomly assigned to a control (CG) or an intervention group (IG). The IG received a short video showing an example of blended therapy, the CG an attention placebo video. Both groups received a reliable online questionnaire assessing acceptance, effort expectancy, performance expectancy, facilitating conditions, social influence and internet anxiety. Between group differences were examined using t-tests and Mann-Whitney tests. Exploratory analysis was conducted to identify moderators. Psychotherapists in CG showed mixed baseline acceptance towards blended therapy (low = 40%, moderate = 33%, high = 27%). IG showed significantly higher acceptance compared to CG ($d$ = .27, $p_{one-sided}$ = .029; low = 24%, moderate = 47%, high = 30%). Bootstrapped confidence intervals were overlapping. Performance expectancy ($d$ = .35), effort expectancy ($d$ = .44) and facilitating conditions ($d$ = .28) were significantly increased (p < .05). No effects on social influence and internet anxiety were found (p>.05). Exploratory analysis indicated psychodynamic oriented psychotherapists profiting particularly from the AFI. Blended therapy is a promising approach to improve healthcare. Psychotherapists show mixed acceptance, which might be improvable by AFIs, particularly in subpopulations of initially rather skeptical psychotherapists. Forthcoming studies should extend the present study by shifting focus from attitudes to the impact of different forms of AFIs on uptake.

**Funding:** The authors received no specific funding for this work.

**Competing interests:** The authors have declared that no competing interests exist.

## Introduction

Mental disorders are globally highly prevalent and affect people in all regions worldwide [1–4], accounting for 32.4% of years lived with disability and 13% of disability adjusted life years [5]. Moreover, the pooled relative risk (RR = 2.22, 95%-CI: 2.12–2.33) of mortality among persons with mental disorders is increased by over 120% compared to persons without mental disorders [6].

Several psychotherapeutic and psychopharmacological interventions are effective in the treatment of mental health conditions [7]. However, low perceived need for help and obstructive attitudes towards mental health treatments limit treatment seeking behavior and staying in treatment [8]. Thus, measures to improve peoples´ attitudes towards mental health care are needed to improve peoples´ intention to use mental health services and ultimately make use of the available evidence-based treatment approaches.

This general call for active dissemination of mental health treatments is particularly true for the evidence-based treatment and prevention approach of internet- and mobile-based interventions (IMIs) [9]. IMIs have been shown to be effective and cost-effective across several mental health conditions [10–14]. They frequently have been suggested as one option to increase the dissemination of mental health care in light of advantages over conventional face to face therapies such as being time efficient, being spatial and timely independently usable and allowing a higher degree of anonymity for those who perceive psychotherapy as stigmatizing [15–18]. On the other hand, IMIs require more self-regulation, self-reflection and self-management competencies, which might lead to time pressure or frustration [19] and might come along with communication problems given the lack of visual cues in the therapeutic process [20].

A newer approach called "blended therapy" integrates the advantages of both IMIs and conventional psychotherapy, aiming to combine the best of two worlds [21]. Blended therapy might save clinicians´ time compared to traditional psychotherapy, increase the effectiveness of current state-of the art treatment, or might on the other side lead to lower dropout rates in IMIs [21]. As such, blended therapy is suggested as a promising innovation for the psychotherapeutic setting [22].

A fundamental precondition to implement blended therapy in routine care would be that both patients and psychotherapists are willing to use blended therapy [23–27]. In the field of stand-alone IMIs several studies showed that the baseline acceptance rate of patients towards IMIs is low amongst different clinical target populations [28–32]. Informational material (acceptance facilitating interventions [AFIs]) such as short informational videos, aiming to provide trustworthy information, reduce apprehensions and misconceptions, proofed to be capable of improving patients´ acceptance of IMIs [28,29,31,33,34]. Moreover, patients seem to prefer psychotherapist assisted e-mental health services over stand-alone IMIs [25], suggesting blended therapy as a way to go [35]. Still, improving patients´ acceptance and offering blended therapy approaches to patients might not be sufficient, when psychotherapists´ attitudes towards digital supported therapies are not positive. Previous studies focusing on stand-alone IMIs showed advocating attitudes in psychotherapists [36–38]. However, only a few studies evaluated attitudes towards blended therapy [27,39]. Schuster and colleagues [27] reported evidence for the general acceptance of blended therapy with no preference of psychotherapists' attitudes towards web-based or blended therapy compared to face-to-face therapy [27]. Also Becker and Jensen-Doss [39] reported positive average attitudes towards blended therapy. Psychotherapists are important gate keepers of patients' treatment choice [40]. Hence, increasing the acceptance of psychotherapists towards blended therapy is of utmost importance. Given their effectiveness in patients AFIs might facilitate psychotherapists´

acceptance of digital supported therapies [27–29,31,33,34]. Currently, there is no randomized controlled trail evaluating the effectiveness of an AFI on the acceptance of psychotherapists' towards blend therapy.

Hence, the present studies aimed (1) to investigate the degree of psychotherapists' acceptance towards blended therapy and (2) to examine the effectiveness of an AFI on psychotherapists' acceptance towards blended therapy. In the context of psychotherapists' attitudes towards stand-alone IMIs theoretical orientation (e.g. cognitive behavioral or psychodynamic) is argued to influence the acceptance [36–38]. Thus, the effect of an AFI might be moderated by the theoretical orientation. To further examine this thesis as well as to identify other potential moderators (3) exploratory analyses were conducted.

## Materials and methods

This study was an experimental study with a balanced (1:1) randomization scheme. In cooperation with five German psychotherapy chambers (*Landespsychotherapeutenkammern Baden-Württemberg*, *Schleswig-Holstein*, *Bayern*, *Hessen*, *Hamburg*) and one medical association (Landesärztekammer Hessen) psychotherapists were recruited from November 2016 till February 2017 via e-mail, internet websites and postal mail. Data were collected via *unipark* (https://www.unipark.com/ [last accessed on 18.02.2020]) and unipark's randomization feature was used to allocate survey participants randomly to either the intervention group (IG) or control group (CG). To be included in the study, persons had to be a licensed psychotherapist or a psychotherapist in training. The IG received an AFI video and the CG received a placebo video. Detailed information on the intervention and control video are provided the section "experimental conditions" below. The study design was presented to the ethics committee of Ulm University which deemed this study as ethically uncritical.

About 13,740 psychotherapists were contacted primarily via the email distribution lists of the aforementioned psychotherapy chambers. Moreover, the study was advertised via chambers´ homepages. A total of 513 psychotherapists visited the online survey, 284 started the survey, and 233 were included in per-protocol analyses. A post-hoc power analysis (one-sided Wilcoxon–Mann–Whitney test for independent groups) revealed a power of $1-ß = .64$ to detect the present effect of $d = 0.27$ on the acceptance score between IG and CG at an α-level of 5%.

### Experimental conditions

Intervention group—Acceptance facilitating intervention (AFI): The AFI was a 5-minute video presenting information about blended therapy. The video discusses potential psychotherapists' worries about the use of IMIs [41] and facets of the Unified Theory of Acceptance and Use of Technology (UTAUT) [42] to influence the attitude towards blended therapy positively. In this way, dysfunctional beliefs and worries were challenged and advantages of blended therapy was emphasized. The video was framed by scenes from a F2F-psychotherapy session, showing a role-play scene between a psychotherapist and a patient, with actors acting the parts. The video showed an exemplarily integration of internet-based interventions into psychotherapy. After a short psychotherapy scene, an expert in IMIs (HB) presented various ways in which IMIs could be integrated in psychotherapy. For example, exercises of an internet-based intervention between F2F-sessions or the use of internet-interventions for comorbid disorders were shown. This was accompanied by further information about patients' empowerment, increased self-efficacy and autonomy, efficient use of F2F-sessions, and improvement of healthcare. To further illustrate benefits of blended therapy as well as to reduce worries, an example patient reported about her positive experience with blended therapy (e.g. experienced

flexibility, additional support, simplicity and usability of internet-interventions or data security). The video was developed in cooperation with the School of Advanced Professional Studies, University Ulm and elements of a real internet-intervention against panic attacks (https://www.geton-training.de/Panik.php) were used for illustrative purposes. For further information about the AFI the video script (German) is presented in the supporting information.

Control group—Attention placebo video: The attention placebo video was a video of four minutes with a psychotherapist talking about work load and work burden of psychotherapists. Thus, the video was relevant for psychotherapists, however, without an expected specific impact on psychotherapists attitude towards blended therapy. The placebo video is available under: http://www.kbv.de/html/22421.php [last accessed on 18.02.2020].

## Measures

**Primary outcome.** Acceptance was operationalized based on the Unified Theory of Acceptance and Use of Technology (UTAUT) [42], which emerged from eight different acceptance models: Theory of Reasoned Action (TRA; [43]), Technology Acceptance Model (TAM, [44]), Motivational Model (MM; [45]), Theory of Planned Behavior (TPB; [46]), Combined TAM and TPB (C-TAM-TPB; [47]), Model of PC Utilization (MPCU; [48]), Innovation Diffusion Theory (IDT; [49] and Social Cognitive Theory (SCT; [50]). While the model was initially developed and validated in the work context [42], the questionnaire has been successfully transferred to the medical field in prior studies (eg. [28,29,31]). As in the original questionnaire, all items were rated on 5-point scales with response options ranging from "does not apply at all (1)" to "applies completely (5)". Four items assessed acceptance: 1. Generally, I would consider to test blended therapy, 2. I would use blended therapy regularly, if I had the possibility, 3. I would recommend blended therapy to colleagues and 4. I would NOT use blended therapy (inverted item). Items are summed for a total acceptance score (range: 5–20, mean = 12.5). For the original items of the UTAUT questionnaire see [42]. Reliability was excellent ($\omega_{total}$ = .94). Furthermore, the construct validity of the adopted UTAUT questionnaire was confirmed in a validation study: Confirmatory factor analysis and structural equation modelling for the proposed UTAUT model yielded an excellent fit [51]. Acceptance was assessed after participants watched the intervention (IG) or attention placebo video (CG).

**Secondary outcomes.** Based on UTAUT four key predictors (performance expectancy, effort expectancy, social influence, facilitating conditions) were operationalized. Items were based on the UTAUT model [42]. Similar, to the acceptance items, the original UTAUT items were adopted to the medical setting. In addition, internet anxiety was added as a dimension based on previous studies (eg. [28,29,31]). Furthermore, inverted items were created and added in this study to reduce biasing effects caused by the assessment methodology. As outline above the model fit of the adopted UTAUT questionnaire and model (including internet anxiety) is excellent [51].

Performance expectancy was measured by 8 items (reliability, $\omega_{total}$ = .93), effort expectancy by 6 items ($\omega_{total}$ = .86), social influence by 3 items ($\omega_{total}$ = .76), facilitating conditions by 7 items ($\omega_{total}$ = .80), and Internet anxiety by 3 items ($\omega_{total}$ = .83). (see Table 1 for items). Items are summed for a total score for each predictor. All five predictor scales were assessed after participants watched the intervention (IG) or attention placebo video (CG).

**Sociodemographic data and other variables.** In addition to age and sex, type of psychotherapy license (psychological psychotherapist, child and adolescents psychotherapist, psychiatrist, psychosomatic practitioner, child and adolescents practitioner, other), therapeutic background (behavior psychotherapy, depth psychotherapy, psychodynamic psychotherapy, other), work setting (practice, counseling center, outpatient clinic, inpatient clinic, other),

**Table 1. Questionnaire items for secondary outcomes.**

*Performance expectancy* (8 items):

 1. Blended therapy would improve the effectiveness of my treatments.

 2. Blended therapy could support my work and increase my productivity.

 3. Blended therapy would help my patients generally.

 4. I expect blended therapy would hinder the therapeutic relationship[1].

 5. Patients' needs cannot be sufficiently targeted by blended therapy[1].

 6. My possibilities to react in certain situations are restricted in blended therapy[1].

 7. Blended therapy will not be beneficial for my work, because its development is not practice orientated[1].

 8. I cannot imagine to use blended therapy, because of its danger for the therapeutic work[1].

*Effort expectancy* (6 items):

 1. Use of blended therapy would be simple.

 2. I could handle blended therapy easily.

 3. Use of blended therapy would be easy and comprehensible.

 4. Creating patients' compliance would be difficult[1].

 5. Use of blended therapy would create a higher workload for myself[1].

 6. It would be hard to integrate blended therapy in my work[1].

*Social influence* (3 items):

 1. My colleagues would advise me to use blended therapy.

 2. My supervisor or experienced colleagues would advise me to use blended therapy.

 3. My colleagues would discourage me from using blended therapy[1].

*Facilitating conditions* (7 items):

 1. I would get support, if I encounter technical problems.

 2. I fulfill all technical requirements to use blended therapy.

 3. Blended therapy can cause problems with data and privacy security[1].

 4. I expect additional costs, if I use blended therapy.

 5. I expect additional costs for my patients, if I use blended therapy[1] (inverted item).

 6. Handling of blended therapy would be difficult for my patients[1].

 7. My patients do not fulfill the technical requirements to use blended therapy[1].

*Internet anxiety* (3 items):

 1. The internet has something threatening to me.

 2. I am afraid making an irrevocable mistake while using the internet.

 3. I am very concerned, when I use the internet.

[1] Inverted item

employment (fulltime, part-time, unemployed, other), technology access at work and home (yes/no), frequency of technology use at work and home (5-point scale), expertise using PCs or internet (5-point scales), prior knowledge of blended therapy (5-point scale), experience with blended therapy (5-point scale) were assessed.

## Data analysis

All outcomes were analyzed on a per-protocol basis (PP). Individuals were included in per-protocol analyses (= IG/CG watched the video), if the automated system check, whether the video was played completely, and the self-report check "I watched the video" were positive.

For descriptive purposes, the acceptance scale was split in three categories. Cut-off values were defined by the authors as: low acceptance (acceptance sum score: 5–9), medium acceptance (sum score 10–15) and high acceptance (sum score 15–20). Percentages were calculated

for each category in total and for both groups. Differences in the frequencies were assessed by Chi-square test.

To assess whether acceptance of intervention group differs significantly from the control group's mean acceptance, one-sided t-test was used for mean differences with alpha level set to 5%. The acceptance score in the IG was compared against the acceptance score of the CG. Acceptance was asses after participants watched the intervention and control video, respectively. In presence of non-normally distributed data Mann-Whitney test and bootstrapping were used. The bootstrapped 2.5% and 97.5% quantile of the distribution of the resampled group means were used to identify a group difference. If the groups' quantiles were overlapping no differences were assumed. The normal distribution assumption was tested via Shapiro-Wilk test. Similar to acceptance Mann-Whitney test and bootstrap was used for secondary outcomes (performance expectancy, effort expectancy, social influence, internet anxiety and facilitating conditions), if scales were not normally distributed. Two-sided tests were used for all secondary outcomes.

A linear regression model was used as explorative analysis to identify moderators on the effect of AFI. The variables age, gender, therapeutic background (behavioral psychotherapy, depth psychotherapy, psychodynamic psychotherapy, other), type of license (psychological psychotherapist, child and adolescents psychotherapist, other) and their interactions with group were inserted in an initial model (all variables were effect coded or z-standardized). Group was dummy-coded (1 = IG). The initial model included a total of 20 predictors, inclusive the intercept and all interactions. In a stepwise procedure non-significant predictors were removed from the model until a final model was achieved. Ordinary Least Square (OLS) estimator was employed.

### Missingness

A total of 31 items were used to measure acceptance, effort expectancy, performance expectancy, internet anxiety, social influence and facilitating conditions. Missingness for acceptance score was 2.2%. For effort expectancy, performance expectancy, internet anxiety and facilitating conditions missingness was 4.7%, respectively. Dropout is assumed to be independent of the included variables and missing values. A missing completely at random mechanism (MCAR) [52] was assumed for all six variables. Analyses based on the original data using list-wise exclusion yield similar results as analyses using multiple imputations based on predictive mean matching (m = 20). Since no differences occurred, only results from analyses based on the original data using list-wise exclusion are reported in the present study.

### Results

Of 284 participants 140 were randomly assigned to the intervention group and 144 to the control group. Mean age was 48.6 (*SD* = 11.7) and 59% of all participants were female. Based on the system check 21.4% of the participants of the intervention group and 7.8% in the control group did not watch the video completely. Accordingly, for the per protocol analyses 107 participants remain as IG and 126 as CG (see Fig 1). Further demographics of the analyzed sample are summarized in Table 2.

### Level of acceptance

Acceptance measured in CG was low to moderate (M = 11.4, *SD* = 4.8; low = 39.8%, moderate = 33.3%, high = 26.8%). Acceptance in IG was moderate to high (M = 12.7, *SD* = 4.5; low = 23.8%, medium = 46.7%, high = 29.5%) (Fig 2). Chi-square test revealed significant differences in the frequencies of acceptance categories ($\chi$2 (2, N = 228) = 7.18, p = 0.028).

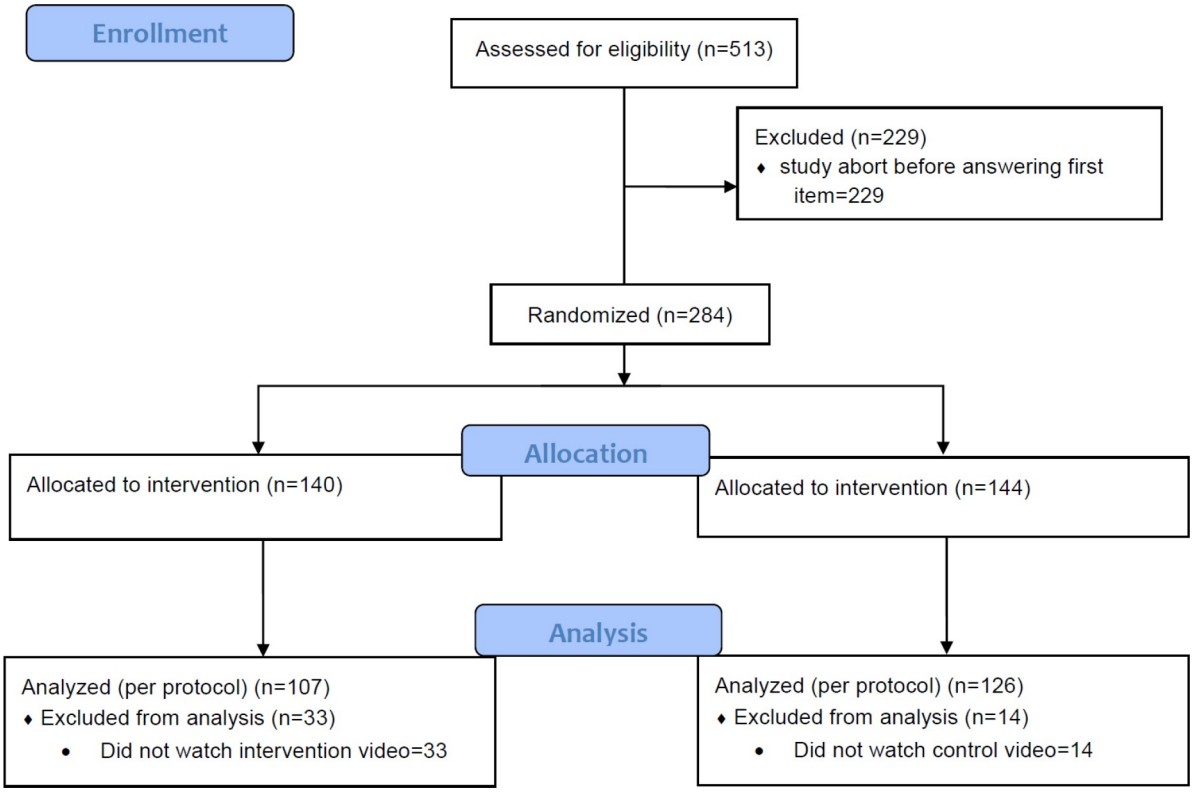

**Fig 1. Flow-chart according to the consort statement.**

## Between-group effect on acceptance

*Acceptance* was not normally distributed (Shapiro-Wilk test: $p < .001$). One-sided Mann-Whitney test showed significant differences between groups ($p_{\text{one-sided}} = .026$). IG showed higher acceptance ($M_{CG} = 11.5$ [95%-*CI*: 10.6–12.3], $M_{IG} = 12.7$ [95%-*CI*: 11.9–13.6]). The difference between CG and IG in standard deviations is $d = 0.27$ (95% CI: .01-.53).

## Between-group effects on secondary outcomes

*Performance expectancy* was not normally distributed (Shapiro-Wilk test: $p < .001$). Mann-Whitney test showed significant differences in location parameters ($p = .011$). The effect in standard deviations is $d = 0.34$ (95%-*CI*: .08–.60) favoring intervention group. Bootstrap resulted in overlapping quantiles ($M_{CG} = 24.0$ [95%-*CI*: 22.6–25.3], $M_{IG} = 26.5$ [95%-*CI*: 25.1–27.9]). *Effort expectancy* was normally distributed (Shapiro-Wilk test: $p = .166$). A t-test revealed significant higher values for IG ($M_{CG} = 17.9$, $M_{IG} = 20.2$, $t(219.93) = -3.51$, $p < .001$). The effect in standard deviations is $d = 0.46$ (95%-*CI*: .20–.71) favoring the IG. *Facilitating conditions* were normally distributed (Shapiro-Wilk test: $p = .130$). T-test was significant ($t(216,58) = -2.00$, $p = .046$). The mean of IG was $M_{IG} = 23.2$ and the mean of CG $M_{CG} = 22.0$ and the effect in standard deviations was $d = 0.27$ (95%-*CI*: .01–.53) favoring intervention group. *Social influence* was not normally distributed (Shapiro-Wilk test: $p < .001$). Mann-Whitney test showed no significant differences in location parameters ($p = .301$) and quantiles of bootstrapped means were overlapping ($M_{CG} = 7.4$ [95%-*CI*: 7.0–7.9], $M_{IG} = 7.87$ [95%-*CI*: 7.41 – 8.34]). *Internet anxiety* was not normally distributed (Shapiro-Wilk test: $p < .001$). Mann-Whitney test showed no significant differences in location parameters ($p = .759$) and

**Table 2. Demographics.**

| | Control group (n = 126) | | Intervention group (n = 107) | |
|---|---|---|---|---|
| | M (SD) \| % | n | M (SD) \| % | n |
| Age | 49.1 (12.9) | | 47.1 (10.8) | |
| Sex | | | | |
| Male | 31.0 | 39 | 17.8 | 19 |
| Female | 52.4 | 66 | 71.0 | 76 |
| Not indicated | 16.6 | 21 | 11.2 | 12 |
| Type of license | | | | |
| Child & adolescents | 26.2 | 33 | 20.6 | 22 |
| Psychological psychotherapist | 63.5 | 80 | 68.2 | 73 |
| Psychosomatic practitioner | 0.01 | 1 | 0 | 0 |
| Other | 3.2 | 4 | 8.4 | 9 |
| Therapeutic background[1] | | | | |
| Behavioral therapy | 54.0 | 68 | 57.0 | 61 |
| Psychodynamic therapy | 13.5 | 17 | 9.3 | 10 |
| Depth psychology | 28.6 | 36 | 28.0 | 6 |
| other | 14.3 | 18 | 14 | 15 |
| Prior-knowledge about blended therapy | | | | |
| Extent | 1.7 (1.1) | | 1.7 (1.1) | |
| Valence | 1.6 (1.8) | | 1.3 (1.8) | |
| Experiences with blended therapy | | | | |
| Extent | 2.3 (1.2) | | 2.3 (1.3) | |
| valence | 2.6 (1.5) | | 2.3 (1.7) | |

[1] multiple choice.

quantiles of bootstrapped means were overlapping ($M_{CG}$ = 5.5 [95%-$CI$: 5.1–6.0], $M_{IG}$ = 5.3 [95%-$CI$: 4.9–5.8]). Effects on secondary outcomes are summarized in Table 3.

### Exploratory analysis of effect moderating variables

Results of the exploratory analysis are summarized in Table 4.

Psychotherapists using depth psychology and other approaches showed less acceptance towards blended therapy compared to psychotherapists with other type of therapeutic background (see Table 1). Individuals within the IG had an increased acceptance by 0.28 standard deviations compared to average. This effect was increased by 0.26 standard deviations for psychodynamic psychotherapists (see Table 1 and Fig 3).

## Discussion

This is the first study examining the effectiveness of an acceptance facilitating intervention [AFI] on the acceptance of psychotherapists towards blended therapy. The effect on the primary outcome acceptance was small to medium ($d$ = 0.27). Further small to medium effects on performance expectancy, effort expectancy and on facilitating conditions were found, but no effects were observed on social influence and internet anxiety. Overall, most psychotherapists showed a moderate to high acceptance in the CG, with an average acceptance substantially above the scale mean. Given the placebo video had no effect on the acceptance, we assume that the acceptance in the CG represents the acceptance in the general psychotherapist population. Explorative analysis revealed that AFI effect on acceptance is almost doubled for

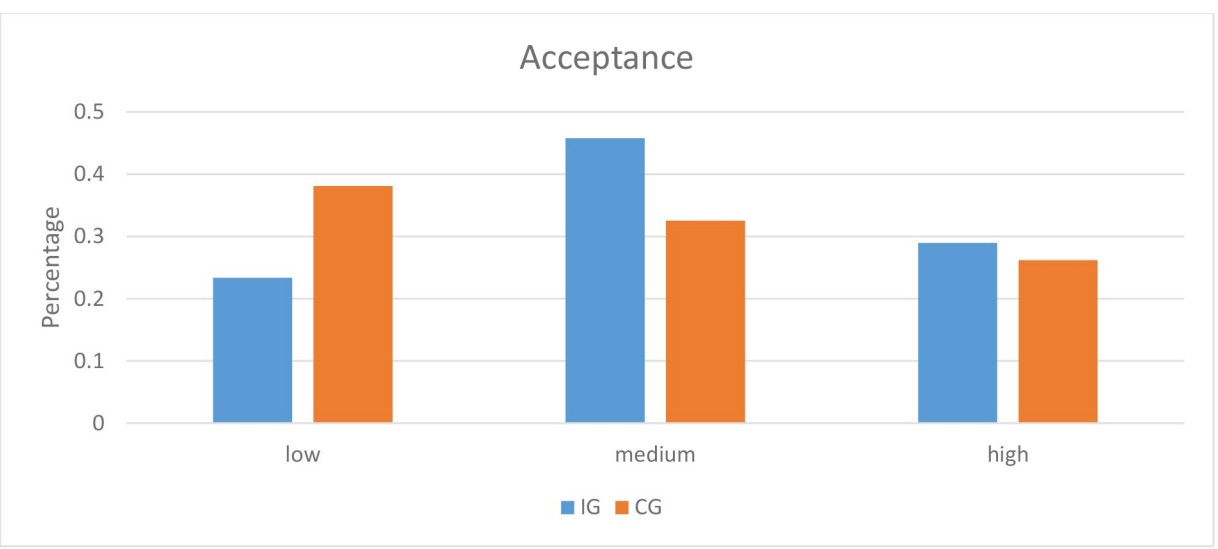

Caption: IG = intervention group; CG = control group

**Fig 2. Acceptance of psychotherapists towards blended therapy in relation to the experimental conditions.**

psychodynamic psychotherapists. Further, explorative analysis showed that the subpopulation of psychotherapist from a depth psychology background had lower than average acceptance.

Blended therapy is a new approach, which combines advantages of both face-to-face psychotherapy and IMIs [21,26] and seems to be a promising approach to improve the current mental health care situation. Based on the above average baseline acceptance found in this study the fundamental precondition for the implementation of online elements into on-site psychotherapy, namely psychotherapists´ acceptance towards blended therapy, seems to be given, which is in line with previous findings [27]. However, the overall acceptance towards blended therapy should be interpreted carefully, as the present study sample is not representative with only a minimal percentage of the psychotherapists following the study invitation, most probably with a bias towards digitally open psychotherapists. The AFI used in this study showed positive effects on psychotherapists´ acceptance towards blended therapy. This result is in line with other studies examining the effects of AFIs on patients' acceptance (e.g. [28,29,31]). However, prior studies found higher effect sizes (e.g. $d = 0.71$ [31]), while showing lower baseline acceptance (e.g. 93.7% of all participants reported a low to moderate acceptance [31]). Based on these differences, one could assume that AFIs are especially effective in populations with low acceptance. The present study design is not able to test this assumption, since a

**Table 3. Effects on secondary outcomes.**

|  | **p-value** | **Effect size** |
| --- | --- | --- |
| Performance expectancy | .011[1] | 0.34 (95 %-*CI*: .08 - .60) |
| Effort expectancy | < .001[2] | 0.46 (95 %-*CI*: .20 - .71) |
| Facilitating conditions | .046[2] | 0.27 (95 %-*CI*: .01 - .53) |
| Social influence | .301[1] | - |
| Internet anxiety | .759[1] | - |

[1] based on Mann-Whitney test

[2] based on t-test.

**Table 4. Exploratory regression results for z-standardized acceptance scores.**

| Predictors | Estimate in SD | 95%-CI[2] |
|---|---|---|
| **Main effects[1]** | | |
| Intercept | -0.09 | -0.28 to 0.09 |
| Group | .28 | 0.07 to 0.49 |
| Therapeutic background | | |
| Depth psychology | -0.37 | -0.52 to -0.21 |
| Psychodynamic | -0.07 | -0.31 to 0.17 |
| **Interaction effects[1]** | | |
| Psychodynamic X Group | 0.26 | 0.04 to 0.47 |

Adjusted R2 = .18, F(5,193) = 11.63, p < .001.

[1] Only significant main and interaction effects are listed. All other predictors were eliminated based on non-significance during the step-wise process.

[2] Confidence intervals are based on bootstrap (100,000 draws).

pre-test in the intervention group would have been necessary to estimate an interaction between baseline acceptance and AFI. However, such a design adaptation should be considered carefully, as a pre-post-test design might not be adequate for examining an ultra-short AFI delivered in under 5 minutes [28,29,31]. Furthermore, it has to be highlighted that the present study was not powered to identify a small to medium effect size of d 0.27. Thus, future studies with confirmatory design should replicate the present findings.

Another aim of the study was to identify potential effect moderators. The explorative analysis revealed a subpopulation, which showed lower than average acceptance (psychotherapist using depth psychology). Further, being a psychodynamic psychotherapist was found to be a meaningful moderator, with psychodynamic psychotherapists profiting to a greater extend from the AFI compared to psychotherapists with another background. This finding suggests that short informational videos might be particularly useful as a first step for those who are skeptical or uninformed regarding blended therapy, whereas short AFIs might not be sufficient to further increase acceptance of psychotherapists who are already to some degree open to this approach.

When interpreting the present findings some limitations need to be taken into account. First, some data within this study was not normally distributed. Typically used parametric methods would have led to biased results. Non-parametric tests and bootstrap were used in the analyses to obtain robust results [53–55]. Thus, differences between CG and IG could be detected and further exploratory analysis was feasible with all assumptions met. For the latter the explorative character should be highlighted. No a-priori assumptions about which variables moderate the effect of the AFI were made, except for psychotherapeutic orientation. Moreover, the study was neither designed nor powered to detected moderator effects. In addition, group sizes for main effects in the explorative analyses were highly unequal (e.g. ratio for other approaches was roughly 1:6), which means the present analyses was also highly underpowered to detect main effects (e.g. post-hoc power analysis to detect a main effect of $d = 0.3$ for other approaches yield a power of 33%). Thus, the generalizability of the exploratory analysis should be interpreted carefully and further studies validating the present findings are needed.

Second, in all analyses missing cases were excluded. This procedure leads to unbiased estimates, if missingness is missing completely at random (MCAR) [52]. Yet, this assumption cannot be statistically verified. In the present study, missingness was rather low (< 5%) and a

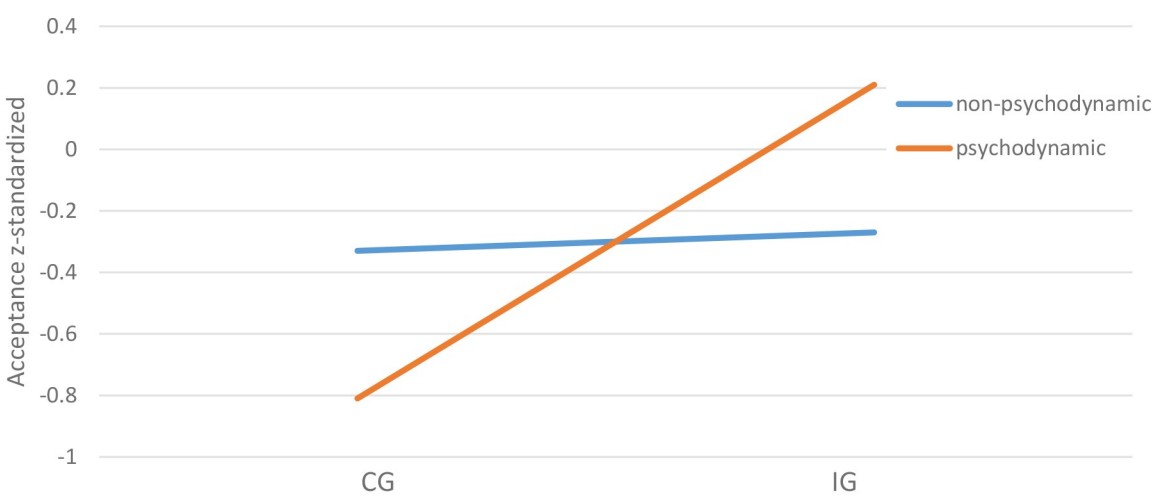

Caption: IG = intervention group; CG = control group; (non-)psychodynamic = theoretical background of psychotherapists

**Fig 3. Interaction effect between effect coded group and effect coded psychodynamic therapist.**

replication with multiple imputations yielded no different results. Hence, results are expected to be at low risk of bias.

Third, there might be a baseline imbalance between the experimental conditions regarding gender, which could have biased the present findings. As the more important it seems to replicate the present findings in further experimental studies in order to substantiate the present evidence.

Fourth, only 284 from potentially over 13,000 psychotherapists took part in the present survey and only 233 were included in analyses. Moreover, participants were recruited mainly via online ways (e.g. emails or website) and the survey was also online. This may have led to a selective "internet friendly/familiar" sample, which is not representative for all psychotherapists. This could also be an explanation for the rather high level of acceptance compared to prior studies in representative patient samples [28,29,31]. At the same time, this argues for a higher impact of AFIs in the whole target group, given the findings, that AFIs were seemingly more effective in initially more skeptical participants.

Fifth, in this study the UTAUT model [42] was used and the AFI and the scales were developed based on this model, extended by internet anxiety as a predictor. All outcomes were measured reliably according to typical cut-off values for internal consistency [56]. Reliability scores were calculated using McDonald's Omega, which is argued to be a better estimator than the often used Cronbach's alpha [57,58]. Although the UTAUT model was used in this study, the UTAUT model itself and the legitimacy of the introduced predictor internet anxiety was not evaluated. Future studies should test whether the UTAUT model and its extension applies for blended therapy.

Finally, a different design including a pre-test (with substantial time between pre- and post-assessment to avoid a retest/recall bias) to test whether AFIs are indeed more effective in low-acceptance population should be applied. Thereby, different AFI designs (e.g. information paper, presentation format, testing of an example online-component, targeting special population characteristics such as therapeutic background) might facilitate acceptance in different ways and should be tested accordingly in order to examine the most efficient way of increasing

participants´ acceptance. For optimization purposes a multiphase optimization strategy (MOST) using fractional factorial designs as recommended by Collins and colleagues could be used [59,60].

## Conclusion

Currently psychotherapists in Germany show a mixed acceptance towards blended therapy, which will likely be in a similar vein found in other countries that are not yet much familiar with digital approaches for treating mental disorders. The AFI within this study had a significant small to moderate overall effect on psychotherapists´ acceptance. Thus, AFIs might be an easy to distribute way of facilitating psychotherapists´ attitudes towards blended therapy. Given that not all participants watched the video, implementation strategies should be developed which ensure that psychotherapists´ actually do watch the video (e.g. as controlled CME training). Finally, forthcoming studies need to go beyond acceptance as the outcome and examine psychotherapists´ actual use of blended therapies, which might shift the focus from attitudes to practical, legal and monetary aspect of implementing blended therapies in our daily psychotherapeutic work. Aiming at increasing actual use of blended therapy, we also expect that AFIs need to be more complex than a five-minute video, using a longitudinal approach based on interventions ranging from information over initial workshops to continuous training on the job.

## Supporting information

**S1 File. Video script.**
(DOCX)

**S2 File. Dataset.**
(XLSX)

## Author Contributions

**Conceptualization:** Harald Baumeister, Rüdiger Nübling, David Daniel Ebert.

**Formal analysis:** Yannik Terhorst.

**Investigation:** Harald Baumeister, Cora Grässle, Maren Freudenstein.

**Methodology:** Harald Baumeister, Yannik Terhorst, Cora Grässle, Maren Freudenstein.

**Project administration:** Harald Baumeister, Cora Grässle, Maren Freudenstein.

**Resources:** Harald Baumeister, Rüdiger Nübling.

**Supervision:** Harald Baumeister, Rüdiger Nübling, David Daniel Ebert.

**Validation:** Harald Baumeister, Rüdiger Nübling, David Daniel Ebert.

**Writing – original draft:** Harald Baumeister.

**Writing – review & editing:** Harald Baumeister, Yannik Terhorst, Cora Grässle, Maren Freudenstein, Rüdiger Nübling, David Daniel Ebert.

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
