## [Decision Letter · Decision Letter 0]

24 Jan 2020

PONE-D-20-00142

Impact of an Acceptance Facilitating Intervention on Psychotherapists’ Acceptance of Blended Psychotherapy

PLOS ONE

Dear Dr. Baumeister,

Thank you for submitting your manuscript to PLOS ONE. After careful consideration, we feel that it has merit but does not fully meet PLOS ONE’s publication criteria as it currently stands. Therefore, we invite you to submit a revised version of the manuscript that addresses the points raised during the review process.

We would appreciate receiving your revised manuscript by February 24, 2020. To enhance the reproducibility of your results, we recommend that if applicable you deposit your laboratory protocols in protocols.io, where a protocol can be assigned its own identifier (DOI) such that it can be cited independently in the future. For instructions see: http://journals.plos.org/plosone/s/submission-guidelines#loc-laboratory-protocols

We look forward to receiving your revised manuscript.

Kind regards,

Stephan Doering, M.D.

Academic Editor

PLOS ONE

3. Please include a caption for figure 3.

Reviewers' comments:

Reviewer's Responses to Questions

**Comments to the Author**

1. Is the manuscript technically sound, and do the data support the conclusions?

Reviewer #1: Yes

Reviewer #2: Partly

2. Has the statistical analysis been performed appropriately and rigorously? 

Reviewer #1: Yes

Reviewer #2: No

3. Have the authors made all data underlying the findings in their manuscript fully available?

Reviewer #1: Yes

Reviewer #2: No

4. Is the manuscript presented in an intelligible fashion and written in standard English?

Reviewer #1: Yes

Reviewer #2: Yes

5. Review Comments to the Author

Reviewer #1: This is an informative paper that addresses an important aspect of psychotherapeutical practice. The authors investigate how an acceptance-facilitating intervention affects psychotherapists' acceptance of blended therapy. Since internet- and mobile-based interventions are becoming an increasingly important element of psychotherapy, this research is highly relevant. The methodology is sound, What is missing is a brief discussion of the current state of research on the topic. The authors claim that their study is the first on acceptance-facilitating interventions regarding psychotherapists' acceptance of blended therapy. However, there are several studies on the attitudes vis-a-vis blended therapy and the acceptance thereof among psychotherapists (e.g. Schuster et al. 2018: https://www.jmir.org/2018/12/e11007/; Mayer et al. 2019: https://mental.jmir.org/2019/11/e14018/). Although the study presented by the authors is methodologically more sophisticated than many of the existing studies, which are mainly survey-based, the authors should mention the existing research on the topic. This helps the reader to better understand the relevance of the paper and it also shows the innovativeness of the study. A few short sentences and two or three sources will suffice.

As a minor point, I would suggest a language check by a native speaker. All in all, the language and style are appropriate, but at some points, the wording is odd or wrong (e.g. "normal distributed" instead of "normally distributed").

Reviewer #2: This is an interesting paper about the integration of technology with the traditional face-to-face way to work in psychotherapy. It could be an initial point to understand this phenomenon. The reduced psychotherapist sample and the absence of a pre-post analysis are the main problems of the research. Additionally there are different parts of the research needed of some explanation to improve the understanding of the paper.

The abstract inform that one of the aims of the study was to identify potential effect moderators. The analysis of moderators is more complex than a regression analysis. The use of this term could be generate confusion. They informed the use of an online questionnaire. It is no clear if is an ad hoc questionnaire or a well know questionnaire with an adequate validity and reliability.

In the introduction paragraph (line 55), they uses the acronym RR with no reference to his meaning.

There is a contradiction between the content of IG video. It is a video showing case (line 108) or a presenting information about blended psychotherapy (line 121)? How is produce the video: with real psychotherapist, actors, etc.? The video is not accessible to the URL provided. Nevertheless, the IG video has a positive orientation and the placebo video a clear negative orientation. Could these aspects affect to the results?

The primary outcome is about “blended psychotherapy”. Is it a term well known by the German psychotherapist?

There are a difference in acceptance baseline between CG and IG groups (line 226-229). A t-test show a significant difference t= 2.131, 231 d.f. p>.025. If there are a difference between groups before the experimental manipulation, the results could be reconsidered.

The authors use the title “effect on acceptance” (line 235) but this term is confused. If there is not a pre-post analysis is not correct to use the term “effect”. This confusion is also present in the “effect on secondary outcomes” paragraph.

There is not specific information about the number of variables introducing in the regression analysis. It is important because the higher the number, the higher the variance explained, but there are a specific ratio between variables and cases.

6. PLOS authors have the option to publish the peer review history of their article (what does this mean?). If published, this will include your full peer review and any attached files.

Reviewer #1: Yes: Dr. Giovanni Rubeis

Reviewer #2: No

---

## [Author Response · Author response to Decision Letter 0]

11 Mar 2020

Please see uploaded point to point reply attached

---

## [Decision Letter · Decision Letter 1]

29 Apr 2020

PONE-D-20-00142R1

Impact of an acceptance facilitating intervention on psychotherapists’ acceptance of blended psychotherapy

PLOS ONE

Dear Dr. Baumeister,

Thank you for submitting your manuscript to PLOS ONE. After careful consideration, we feel that it has merit but does not fully meet PLOS ONE’s publication criteria as it currently stands. Therefore, we invite you to submit a revised version of the manuscript that addresses the points raised during the review process. Whiule reviewer 1 was satisfied with the revision of your manuscript, reviewer 3raised a few minor issues. Please adress these in your revision.

We would appreciate receiving your revised manuscript by May 29, 2020. To enhance the reproducibility of your results, we recommend that if applicable you deposit your laboratory protocols in protocols.io, where a protocol can be assigned its own identifier (DOI) such that it can be cited independently in the future. For instructions see: http://journals.plos.org/plosone/s/submission-guidelines#loc-laboratory-protocols

We look forward to receiving your revised manuscript.

Kind regards,

Stephan Doering, M.D.

Academic Editor

PLOS ONE

Reviewers' comments:

Reviewer's Responses to Questions

**Comments to the Author**

1. If the authors have adequately addressed your comments raised in a previous round of review and you feel that this manuscript is now acceptable for publication, you may indicate that here to bypass the “Comments to the Author” section, enter your conflict of interest statement in the “Confidential to Editor” section, and submit your "Accept" recommendation.

Reviewer #1: All comments have been addressed

Reviewer #3: (No Response)

2. Is the manuscript technically sound, and do the data support the conclusions?

Reviewer #1: Yes

Reviewer #3: Yes

3. Has the statistical analysis been performed appropriately and rigorously? 

Reviewer #1: Yes

Reviewer #3: Yes

4. Have the authors made all data underlying the findings in their manuscript fully available?

Reviewer #1: Yes

Reviewer #3: Yes

5. Is the manuscript presented in an intelligible fashion and written in standard English?

Reviewer #1: (No Response)

Reviewer #3: Yes

6. Review Comments to the Author

Reviewer #1: All comments have been addressed by the authors. The paper is acceptable for publication in its present form

Reviewer #3: This is an interesting article to read, pointing to the importance of acceptance for blended psychotherapy in the treatment giver rather than receiver. Authors have made an effort to improve the manuscript by carefully considering previous reviewers comments. However, there are some points that I think still needs to be addressed.

1. Measures. The questionnaire used in this article, as well as its origins from UTAUT should be more thoroughly described in the method section. Readers who are unfamiliar with this instrument should be provided with enough information to evaluate the applicability of it. For instance, it would be helpful if you added more details on the original instrument, such as total amount of items and number of factors. Please describe how items (or factors) were chosen from the UTAUT, and in what manner they were adapted from the original version (such as; if items are rephrased completely, or adapted by changing some words in the items, and why inverted items were added). If your version is currently being evaluated for psychometric properties, it would be helpful if you mention this in the manuscript. Further, it is unclear where the items on internet anxiety were derived from; the UTAUT or developed by authors?

Since the result section describes deviance from the scale mean, the scale mean should also be stated here (or in the result section if you are referring to the mean of the study sample).

2. Materials and methods. It should be more clearly stated when data was collected, this will make the result section easier to understand, that results are exploring differences after manipulation, not baseline. (You provided this clearly in your answer to reviewer 2, however it is not as clearly described in the article).

3. Data analysis. Authors describe splitting acceptance in three categories (low/medium/high), for descriptive purposes. Could you clarify how you decided on these cut-offs? Are they based on the distribution of responses? Or theoretically derived? Is there any reason not to analyze difference in distributions on acceptance statistically? I think that a chi-square test would be informative on difference in level of acceptance. (seems to be a typo in the medium range where I suppose the range is above 9 and below 16? line 189)

4. In the result section you start by describing a rather high dropout rate in the intervention group. I think the article would benefit from an analysis on difference in characteristics between those who dropped out and those who were completers. This would give a hint if there is a bias in the study sample, if those who dropped out were mostly men, mostly from a specific therapeutic background and so on.

5. In Table 2 it seems as if there is a difference in distribution of gender between groups, with a higher ratio of women to men in the intervention group as compared to control group. Have you checked if this (or other demographics) is a statistically significant difference in distribution? Thoughts on implications if so? (In my version the tab-spacing made table 2 almost uninterpretable, be sure to double-check this before publication).

6. The concept of blended psychotherapy and blended therapy are used interchangeably throughout the manuscript (also blend therapy, line 93). I think it would enhance the readability if authors clarify what they refer to with both concepts, or to stick with only one of them.

7. Also as a minor point on the fluency of language; the sentence in line 75 – 79 I would consider rephrasing. Line 89-90 “it seems…” As a reader I am confused as to what “it” refers to. It is probably better to use AFIs/blended psychotherapy/ psychotherapists.

7. PLOS authors have the option to publish the peer review history of their article (what does this mean?). If published, this will include your full peer review and any attached files.

Reviewer #1: Yes: Dr. Giovanni Rubeis

Reviewer #3: Yes: Maria Fogelkvist

---

## [Author Response · Author response to Decision Letter 1]

25 May 2020

Point-to-point reply to reviewers´ comments on our revised manuscript:

“Impact of an acceptance facilitating intervention on psychotherapists’ acceptance of blended psychotherapy”

Changes in the revised manuscript are highlighted. 

Reviewer #1: All comments have been addressed by the authors. The paper is acceptable for publication in its present form

Note: Thanks for your time and your helpful comments! 

Reviewer #3: This is an interesting article to read, pointing to the importance of acceptance for blended psychotherapy in the treatment giver rather than receiver. Authors have made an effort to improve the manuscript by carefully considering previous reviewers comments. However, there are some points that I think still needs to be addressed.

1. Measures. The questionnaire used in this article, as well as its origins from UTAUT should be more thoroughly described in the method section. Readers who are unfamiliar with this instrument should be provided with enough information to evaluate the applicability of it. For instance, it would be helpful if you added more details on the original instrument, such as total amount of items and number of factors. Please describe how items (or factors) were chosen from the UTAUT, and in what manner they were adapted from the original version (such as; if items are rephrased completely, or adapted by changing some words in the items, and why inverted items were added). If your version is currently being evaluated for psychometric properties, it would be helpful if you mention this in the manuscript. Further, it is unclear where the items on internet anxiety were derived from; the UTAUT or developed by authors?

Note: Added to the manuscript (see amendments to “Measures – paragraphs “Primary outcome” and “secondary outcome”). 

Since the result section describes deviance from the scale mean, the scale mean should also be stated here (or in the result section if you are referring to the mean of the study sample).

Note: Added to the manuscript (mean and range)

2. Materials and methods. It should be more clearly stated when data was collected, this will make the result section easier to understand, that results are exploring differences after manipulation, not baseline. (You provided this clearly in your answer to reviewer 2, however it is not as clearly described in the article).

Note: Added to the methods section (p. 8 “Acceptance was assessed after participants watched the intervention (IG) or attention placebo video (CG).”)

3. Data analysis. Authors describe splitting acceptance in three categories (low/medium/high), for descriptive purposes. Could you clarify how you decided on these cut-offs? Are they based on the distribution of responses? Or theoretically derived? Is there any reason not to analyze difference in distributions on acceptance statistically? I think that a chi-square test would be informative on difference in level of acceptance. (seems to be a typo in the medium range where I suppose the range is above 9 and below 16? line 189)

Note: we added chi-square. The authors, corresponding to prior publications on this scale, have defined the cut-off values based on a theoretical basis. We added this information to the manuscript. 

4. In the result section you start by describing a rather high dropout rate in the intervention group. I think the article would benefit from an analysis on difference in characteristics between those who dropped out and those who were completers. This would give a hint if there is a bias in the study sample, if those who dropped out were mostly men, mostly from a specific therapeutic background and so on.

Note: We additionally ran all analysis based on multiple imputed data. For the imputation models we followed the recommendations by van Buuren and Groothuis-Oudshoorn (2011). Imputation models included: predictors included based on theoretical considerations, correlations between variables and correlations with non-response. Given this procedure gender or therapeutic background (or any other variable) which is associated with drop-out was included in the imputation. Since, the analysis using multiple imputations yielded no difference a systematic bias/ influence is unlikely. 

van Buuren S, Groothuis-Oudshoorn K. mice : Multivariate Imputation by Chained Equations in R. J Stat Softw [Internet]. 2011;45(3). 

5. In Table 2 it seems as if there is a difference in distribution of gender between groups, with a higher ratio of women to men in the intervention group as compared to control group. Have you checked if this (or other demographics) is a statistically significant difference in distribution? Thoughts on implications if so? (In my version the tab-spacing made table 2 almost uninterpretable, be sure to double-check this before publication).

Note: We did not run statistical tests examining potential baseline imbalance, thereby following CONSORT statement recommendations (Moher et al. 2010). However, we agree that there might be a gender imbalance, hence, we added this potential risk of bias to the limitation section of our manuscript. 

Moher et al. ConSoRT 2010 explanation and elaboration: updated guidelines for reporting parallel group randomised trials. BMJ 2010;340:c869

p. 17: “Third, there might be a baseline imbalance between the experimental conditions regarding gender, which could have biased the present findings. As the more important it seems to replicate the present findings in further experimental studies in order to substantiate the present evidence.”

6. The concept of blended psychotherapy and blended therapy are used interchangeably throughout the manuscript (also blend therapy, line 93). I think it would enhance the readability if authors clarify what they refer to with both concepts, or to stick with only one of them.

Note: we now use the term “blended therapy” throughout the manuscript. 

7. Also as a minor point on the fluency of language; the sentence in line 75 – 79 I would consider rephrasing. Line 89-90 “it seems…” As a reader I am confused as to what “it” refers to. It is probably better to use AFIs/blended psychotherapy/ psychotherapists.

Note: Changed, thanks. 

Thanks to all reviewers for their valuable time and insights!

---

## [Editor Report · Decision Letter 2]

11 Jun 2020

PONE-D-20-00142R2

Impact of an acceptance facilitating intervention on psychotherapists’ acceptance of blended therapy

PLOS ONE

Dear Dr. Baumeister,

Thank you for submitting your manuscript to PLOS ONE. After careful consideration, we feel that your manuscript is very cloce to acceptance. Since reviewer 1 is completely satisfied with you revision of the previous version of ypou manuscript, reviewer 3 raises some minor concerns. I assume, that it will not cause too much effort to include these few changes. Therefore, we invite you to submit another revised version of the manuscript that addresses the points raised during the review process.

We look forward to receiving your revised manuscript.

Kind regards,

Stephan Doering, M.D.

Academic Editor

PLOS ONE

---

## [Author Response · Author response to Decision Letter 2]

16 Jul 2020

Dear Editors,

thanks for inviting us to revise our manuscript. I just can´t find the reviewer 3 minor comments. There was no link and no attachment and I can´t find any information in the online submission system. What do I miss here?

Best wishes,

Harald Baumeister

---

## [Editor Report · Decision Letter 3]

20 Jul 2020

Impact of an acceptance facilitating intervention on psychotherapists’ acceptance of blended therapy

PONE-D-20-00142R3

Dear Dr. Baumeister

We’re pleased to inform you that your manuscript has been judged scientifically suitable for publication and will be formally accepted for publication once it meets all outstanding technical requirements.

Kind regards,

Stephan Doering, M.D.

Academic Editor

PLOS ONE

---

## [Editor Report · Acceptance letter]

24 Jul 2020

PONE-D-20-00142R3 

Impact of an acceptance facilitating intervention on psychotherapists’ acceptance of blended therapy 

Dear Dr. Baumeister:

I'm pleased to inform you that your manuscript has been deemed suitable for publication in PLOS ONE. Congratulations! Your manuscript is now with our production department. 

Kind regards, 

on behalf of

Professor Stephan Doering 

Academic Editor

PLOS ONE